# LEARNING TO PERCEIVE OBJECTS BY PREDICTION

## ABSTRACT

The representation of objects is the building block of higher-level concepts. Infants develop the notion of objects without supervision, for which the prediction error of future sensory input is likely a major teaching signal. We assume that the goal of representing objects distinctly is to allow the prediction of the coherent motion of all parts of an object independently from the background while keeping track of relatively fewer parameters of the object's motion. To realize this, we propose a framework to extract object-centric representations from single 2D images by learning to predict future scenes containing moving objects. The model learns to explicitly infer objects' locations in a 3D environment, generate 2D segmentation masks of objects, and perceive depth. Importantly, the model requires no supervision or pre-training but assumes rigid-body motion and only needs observer's self-motion at training time. Further, by evaluating on a new synthetic dataset with more complex textures of objects and background, we found our model overcomes the reliance on clustering colors for segmenting objects, which is a limitation for previous models not using motion information. Our work demonstrates a new approach to learning symbolic representation grounded in sensation and action.

## 1 INTRODUCTION

Visual scenes are composed of various objects in front of backgrounds. Discovering objects from 2D images and inferring their 3D locations is crucial for planning actions in robotics (Devin et al., 2018; Wang et al., 2019) and this can potentially provide better abstraction of the environment for reinforcement learning (RL), e.g. Veerapaneni et al. (2020). The appearance and spatial arrangement of objects, together with the lighting and the viewing angle, determine the 2D images formed on the retina or a camera. Therefore, objects are latent causes of 2D images, and discovering objects is a process of inferring latent causes (Kersten et al., 2004). The predominant approach in computer vision for identifying and localizing objects rely on supervised learning to infer bounding boxes (Ren et al., 2015; Redmon et al., 2016) or pixel-level segmentation of objects (Chen et al., 2017). However, the supervised approach requires expensive human labeling. It is also difficult to label every possible category of objects. Therefore, an increasing interest has developed recently in the domain of *object-centric representation learning* (OCRL) to build unsupervised or self-supervised models to infer objects from images, such as MONet (Burgess et al., 2019), IODINE (Greff et al., 2019) slot-attention (Locatello et al., 2020), GENESIS (Engelcke et al., 2019; 2021), C-SWM (Kipf et al., 2019), mulMON (Nanbo et al., 2020) and SAVi++(Elsayed et al., 2022).

The majority of the early OCRL works are demonstrated on relatively simple scenes with objects of pure colors and background lacking complex textures. As recently pointed out, the success of several recent models based on a variational auto-encoder (VAE) architecture (Kingma & Welling, 2013; Rezende et al., 2014) depends on a capacity bottleneck that needs to be intricately balanced against a reconstruction loss(Engelcke et al., 2020). Potentially due to the lack of sufficient inductive bias existing in real-world environments, such methods often fail in scenes with complex textures on objects and background (Greff et al., 2019). To overcome this limitation, recent works utilized optical flow (either ground truth or estimated) as prediction target (Kipf et al., 2021) because optical flow is often coherent within objects and distinct from the background. Similarly, depth often exhibits sharp changes across object boundaries. When used as an additional prediction target, it further improves segmentation performance of the slot-attention model (Elsayed et al., 2022). Although these new prediction targets allow models to perform better in unsupervised object segmentation in realistic environments with complex textures, these models still pose sharp contrast to the learning

ability of the brain: neither depth nor optical flow are available as external input to the brain, yet infants can learn to understand the concept of object by 8 months old (Piaget & Cook, 1952; Flavell, 1963), with other evidence suggesting this may be achieved as early as 3.5-4.5 months (Baillargeon, 1987). The fact that this ability develops before they can name objects (around 12 months old) without supervision confirms the importance of learning object-centric representation for developing higher-level concepts and the gap of current models from the brain. To narrow this gap, this paper starts with considering the constraints faced by the brain and proposes a new architecture and learning objective using signals similar to what the brain has access to.

As the brain lacks direct external supervision for object segmentation, the most likely learning signal is from the error of predicting the future. In the brain, a copy of the motor command (efference copy) is sent from the motor cortex simultaneously to the sensory cortex, which is hypothesized to facilitate the prediction of changes in sensory input due to self-generated motion (Feinberg, 1978). What remains to be predicted are changes in visual input due to the motion of external objects. Therefore, we assume that the functional purpose of grouping pixels into objects is to allow the prediction of the motion of the constituting pixels in an object in a coherent way by tracking very few parameters (e.g., the location, pose, and speed of an object). Driven by this hypothesis, our contribution in this paper is: (1) we combine predictive learning and explicit 3D motion prediction to learn 3D-aware object-centric representation from RGB image input without any supervision or pre-training, which we call Object Perception by Predictive LEarning (OPPLE); (2) we provide a new dataset[1] with complex surface texture and motion by both the camera and objects to evaluate object-centric representation models; we confirm that several previous models overly rely on clustering colors to segment objects on this dataset; (3) although our model leverages image prediction as a learning objective, the architecture generalizes the ability of object segmentation and spatial localization to single-frame images.

## 2 METHOD

Here, we outline our problem statement then explain details of our model parts, the prediction approach, and the learning objective. Pseudocode for our algorithm and the details of implementation are provided in the appendix (A.1, A.2).

### 2.1 PROBLEM FORMULATION

We denote a scene as a set of distinct objects and a background $\mathbb{S} = \{O_1, O_2, \ldots, O_K, B\}$, where $K$ is the number of objects in scene. At any moment $t$, we denote two state variables, the location and pose of each object from the perspective of an observer (camera), as $\boldsymbol{x}_{1:K}^{(t)}$ and $\boldsymbol{\phi}_{1:K}^{(t)}$, where $\boldsymbol{x}_k^{(t)}$ is the 3-d coordinate of the $k$-th object and $\phi_k^{(t)}$ is its yaw angle from a canonical pose, as viewed from the reference frame of the camera (for simplicity, we do not consider pitch and roll here and leave it for future work to extend to 3D pose). At time $t$, given the location of the camera $\boldsymbol{o}^{(t)} \in \mathbb{R}^3$ and its facing direction $\alpha^{(t)}$, $\mathbb{S}$ renders a 2D image on the camera as $\boldsymbol{I}^{(t)} \in \mathbb{R}^{w \times h \times 3}$, where $w \times h$ is the size of the image. Our goal is to develop a neural network that infers properties of objects given only a single image $\boldsymbol{I}^{(t)}$ as the sole input without external supervision and with only the information of the intrinsics and ego-motion of the camera:

$$\{\boldsymbol{z}_{1:K}^{(t)}, \boldsymbol{\pi}_{1:K+1}^{(t)}, \hat{\boldsymbol{x}}_{1:K}^{(t)}, \boldsymbol{p}_{\phi_{1:K}}^{(t)}\} = f_{\text{obj}}(\boldsymbol{I}^{(t)}) \tag{1}$$

Here, $\boldsymbol{z}_{1:K}^{(t)}$ is a set of view-invariant vectors representing the identity of each object $k$. "View-invariant" is loosely defined as $|\boldsymbol{z}_k^{(t)} - \boldsymbol{z}_k^{(t+\Delta t)}| < |\boldsymbol{z}_k^{(t)} - \boldsymbol{z}_l^{(t)}|$ for $k \neq l$ and $\Delta t > 0$ in most cases, i.e., the vector codes are more similar for the same object across views than they are for different objects. $\boldsymbol{\pi}_{1:K+1}^{(t)} \in \mathbb{R}^{(K+1) \times w \times h}$ are the probabilities that each pixel belongs to any of the objects or the background ($\sum_k \pi_{kij} = 1$ for any pixel at $i, j$), which achieves object segmentation. To localize objects, $\hat{\boldsymbol{x}}_{1:K}^{(t)}$ are the estimated 3D locations of each object relative to the observer and $\boldsymbol{p}_{\phi_{1:K}}^{(t)}$ are the estimated probability distributions of the poses of each object. Each $\boldsymbol{p}_{\phi_k}^{(t)} \in \mathbb{R}^b$ is a probability distribution over $b$ equally-spaced bins of yaw angles in $(0, 2\pi)$.

---

[1] We will release upon publication of the paper

## 2.2 PRINCIPLE BEHIND LEARNING OBJECT REPRESENTATION FROM PREDICTION

Our hypothesis is that the notion of object emerges to meet the need of efficiently predicting the future fates of all parts of an object. With the (to be learned) ability to infer an object's pose and location from each frame, the object's speed of translation and rotation can be estimated from consecutive frames, assuming the camera's self-motion is known. If depth is further inferred for each pixel belonging to an object, then the optical flow of each pixel can be predicted based on the object's speed and the position of each pixel relative to the object's center. The pixel-segmentation of an object essentially prescribes which pixels should move together with the object. With the predicted optical flow, one can further predict part of the next image by warping the current image. The parts of the next image unpredictable by warping include surfaces of objects or the background that are currently occluded but will become visible, and the region of a scene newly entering the view due to self- or object-motion. These portions can only be predicted based on the learned statistics of the appearance of objects and background, which we call "imagination". In this work, we will show that with the information of self-motion, knowledge of geometry (rule of rigid-body movement) and the assumption of smooth object movement, the object representations captured by function $f_{\mathrm{obj}}$ and depth perception can be learned by minimizing the prediction error of the next scene in environments with motion of both the camera and objects, without supervision.

## 2.3 NETWORK ARCHITECTURE

To demonstrate the hypothesized principle above, we build our OPPLE networks as illustrated in Figure 1, which process two consecutive images separately and make prediction for the next image with the information extracted from them.

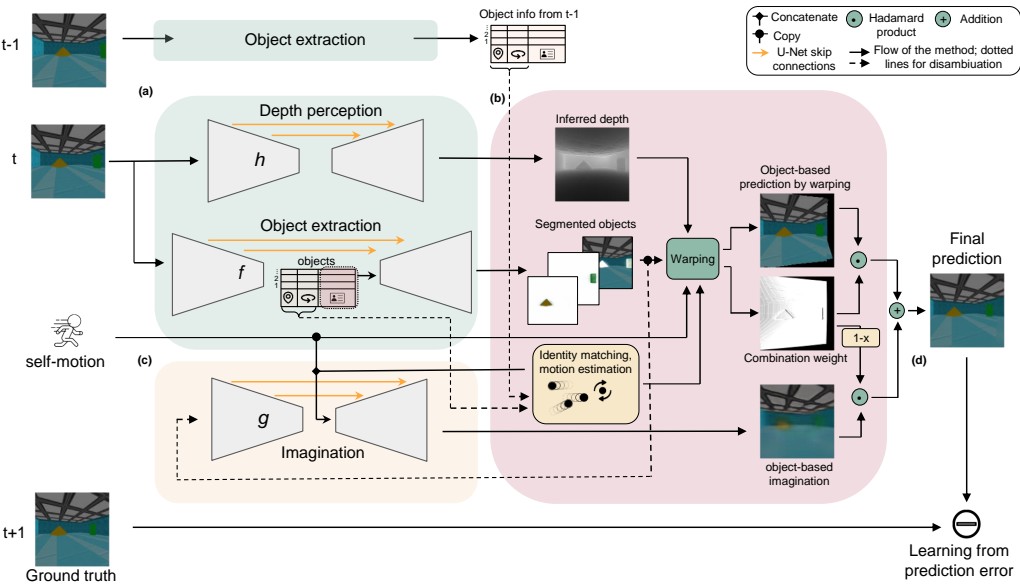

Figure 1: Architecture for the *Object Perception by Predictive LEarning (OPPLE)* network: Our method uses three ordered frames. **(a)** At time $t - 1$, we only extract the object information which gives us the object location and pose. At time frame $t$, we extract depth using a UNet and extract object information using a modified UNet. **(b)** Motion information of each object is estimated from the spatial information extracted for each object at $t - 1$ and $t$. Objects between frames are soft-matched by a score depending on the distance between their latent codes. Self- and object motion information are used together with object segmentation and depth map to predict the next image by warping the current image. **(c)** The segmented object images and depth, together with their motion information and the observer's motion, are used by the imagination network to imagine the next time frame and fill the gap not predictable by warping. **(d)** The error between the final combined prediction and the ground truth of the time frame provides teaching signals for all three networks.

**Depth perception network.** We use a standard U-Net for depth perception function $h_{\theta_{\mathrm{depth}}}$ that processes images $\boldsymbol{I}^{(t)}$ and output a single-channel depth map $\boldsymbol{D}^{(t)}$.

**Object extraction network.** We build an object extraction network $f_{\theta_{\text{obj}}}$ by modificaiton of a U-Net (Ronneberger et al., 2015) to extract representation for each object, the pixels it occupies and its spatial location and pose. A basic U-Net is composed of a convolutional encoder and a transposed convolutional decoder, while each encoder layer sends a skip connection to the corresponding decoder layer, so that the decoder can combine both global and local information. Inside our $f_{\theta_{\text{obj}}}$, an image $\boldsymbol{I}^{(t)}$ first passes through the encoder. Additional Atrous spatial pyramid pooling layer (Chen et al., 2017) is inserted between the middle two convolutional layers of the encoder to expand the receptive field. The top layer of the encoder outputs a feature vector $e^t$ capturing the global information of the scene. A Long-Short Term Memory (LSTM) network further repeatedly reads in $e^{(t)}$ and sequentially outputs one vector for an object at a time. Each object vector is then mapped through a one-layer fully connected network to predict object code $\boldsymbol{z}_k^{(t)}$, object location $\hat{\boldsymbol{x}}_k^{(t)}$ and object pose probability $\boldsymbol{p}_{\phi_k^{(t)}}$, $k = 1, 2, \cdots, K$. The inferred location is restricted within a viewing angle of 1.2 times the field of view and a bounded distance from the camera. The pose prediction is represented as $\log \boldsymbol{p}_{\phi_k^{(t)}}$ for numerical stability. Each object code $\boldsymbol{z}_k^{(t)}$ is then independently fed through the decoder with shared skip connection from the encoder. The decoder outputs one channel for each pixel, representing an un-normalized log probability that the pixel belongs to the object $k$ (a logit map). The unnormalized logit maps for all objects are concatenated with a map of all zero (for the background), and they compete through a softmax function to output the probabilistic segmentation map $\boldsymbol{\pi}_k^{(t)}$ for the probability that each pixel belongs to any of the objects or the background.

**Object-based imagination network.** We build an imagination network $g_{\theta_{\text{imag}}}$ also with a modified U-Net. For each object and the background, the input is concatenated image $\boldsymbol{I}^{(t)}$ and log of depth $log(\boldsymbol{D}^{(t)})$ inferred by the depth perception network, both multiplied element-wise by one probabilistic mask from $\boldsymbol{\pi}_k^{(t)}$. The output of the encoder part of the network is concatenated with a vector composed of the observer's moving velocity $\boldsymbol{v}_{\text{obs}}^{(t)}$ and rotational speed $\boldsymbol{\omega}_{\text{obs}}^{(t)}$, and the estimated object location $\hat{\boldsymbol{x}}_k^{(t)}$, velocity $\hat{\boldsymbol{v}}_k^{(t)}$ and rotational speed $\hat{\boldsymbol{\omega}}_k^{(t)}$ before entering the decoder. The decoder outputs five channels for each pixel: three for predicting RGB colors, one for depth and one for the probability of the pixel belonging to any object $k$ or background. Each predicted RGB and depth image are weighted by the probability image and summed to form the final "imagination". The imagined RGB and depth image for the next frame is further combined with the prediction of the next frame to form the final prediction, detailed in the next section.

## 2.4 PREDICTING OBJECTS' SPATIAL STATES

We start with predicting the spatial states, namely 3D location and pose, of each object for $t + 1$. If the $k^{th}$ object at $t - 1$ is the same as the $k^{th}$ object at $t$ we can use the inferred current and previous locations $\hat{\boldsymbol{x}}_k^{(t)}$ and $\hat{\boldsymbol{x}}_k^{(t-1)}$ for that object to estimate its instantaneous velocity (treating the interval between consecutive frames as the time unit) $\hat{\boldsymbol{v}}_k^{(t)} = \hat{\boldsymbol{x}}_k^{(t)} - \hat{\boldsymbol{x}}_k^{(t-1)}$. Similarly, with the inferred current and previous pose probabilities of the object, $\boldsymbol{p}_{\phi_k}^{(t)}$ and $\boldsymbol{p}_{\phi_k}^{(t-1)}$, we can calculate the likelihood function of its angular velocity $\omega_k^{(t)}$ as $p(\phi_k^{(t)}, \phi_k^{(t-1)} \mid \omega_k^{(t)} = \omega)$. Combining with a Von Mises prior distribution, we can calculate the posterior distribution of the angular velocity $p(\omega_k^{(t)} \mid \phi_k^{(t)}, \phi_k^{(t-1)})$. The location of object $k$ at $t + 1$ can be predicted as $\boldsymbol{x}'_k^{(t+1)} = \boldsymbol{M}_{-\omega_{\text{obs}}}^{(t)}(\hat{\boldsymbol{x}}_k^{(t)} + \hat{\boldsymbol{v}}_k^{(t)} - \boldsymbol{v}_{\text{obs}}^{(t)})$, where $\boldsymbol{v}_{\text{obs}}^{(t)}$ is the velocity of the observer relative to its own reference frame and $\boldsymbol{M}_{-\omega_{\text{obs}}}^{(t)}$ is the rotational matrices due the observer's self rotation. Its new probability for pose being equal to each possible discrete yaw angle bin $\gamma_2$ can be predicted through $p'(\phi_k^{(t+1)} + \omega_{\text{obs}} = \gamma_2) = \sum_{\substack{\gamma_1, \omega, \gamma_2 - \gamma_1 \in \\ \{\omega - 2\pi, \omega, \omega + 2\pi\}}} p(\omega_k^{(t)} = \omega) p(\phi_k^{(t)} = \gamma_1)$, where $\omega_{\text{obs}}$ is the angular velocity of the observer. Since we model $\phi_k^{(t)}$ and $\omega_k^{(t)}$ as probability distributions on discrete bins of angle while $\omega_{\text{obs}}$ can take continuous values, we convert $p'(\phi_k^{(t+1)} + \omega_{\text{obs}})$ to $p'(\phi_k^{(t+1)})$ on the same set of bins as $\phi_k^{(t)}$ by interpolation.

However, there is one important issue: in order to predict the spatial state of each object at $t + 1$ based on the views at $t$ and $t - 1$, the network needs to match the representation of an object at

$t$ from the representation of the same object at $t - 1$. As the dimensions of features (e.g., shape, surface texture, size, etc.) grows, the number of possible objects grows exponentially. Therefore, we cannot simply assume that the $k^{th}$ objects extracted by LSTM from consecutive scenes correspond to the same object, as this requires learning a consistent order over enormous amount of objects. Instead, we take a soft-matching approach: we take a subset (10 dimensions) of the object code in $z_k^{(t)}$ extracted by $f$ as an identity code for each object. For object $k$ at time $t$, we calculate the distance between its identity code and those of all objects at $t - 1$, and pass the distances through a radial basis function to serve as a matching score $r_{kl}$ indicating how closely the object $k$ at $t$ matches each of the objects $l$ at $t - 1$. The scores are used to weight all the translational and angular velocity for object $k$ estimated with the equations above, each assuming a different object $l$ were the true object $k$ at $t - 1$, to yield the final estimated translational and angular velocity for object $k$ at $t$. We additionally introduce a fixed identity code $z_{K+1} = 0$ for the background and set the predicted motion of background to zero.

### 2.4.1 PREDICTING IMAGE BY WARPING

The next image $I^{(t+1)}$ can be partially predicted by warping the current image $I^{(t)}$ based on the predicted optical flow of each pixel in $I^{(t)}$. To predict the optical flow, we need to combine inferred depth of each pixel, inferred probabilities that it belongs to each object and the background, the estimated motion of each object and the knowledge of the camera's self-motion. The model assumes all objects are rigid.

We now consider the fates of all visible pixels belonging to an object. With depth $D^{(t)} \in \mathbb{R}^{w \times h} = h_\theta(I^{(t)})$ of all pixels in a view inferred by the Depth Perception network based on visual features in the image $I^{(t)}$, the 3D location relative to the camera of any pixel $\hat{m}_{(i,j)}^{(t)}$ at coordinate $(i, j)$ in the image can be calculated given the focal length $d$ of the camera. Assuming a pixel $(i, j)$ belongs to object $k$, the estimated motion information $\hat{v}_k^{(t)}$ and $p(\omega_k^{(t)} \mid \phi_k^{(t)}, \phi_k^{(t-1)})$ of the object, together with the current location and pose of the object and the current 3D location $\hat{m}_{(i,j)}^{(t)}$ of the pixel, the 3D location $m'_{k,(i,j)}^{(t+1)}$ of the pixel at the next moment can be predicted as $m'_{k,(i,j)}^{(t+1)} = M_{-\omega_{\mathrm{obs}}}^{(t)}[M_{\hat{\omega}_k}^{(t)}(\hat{m}_{(i,j)}^{(t)} - \hat{x}_k^{(t)}) + \hat{x}_k^{(t)} + \hat{v}_k^{(t)} - v_{\mathrm{obs}}^{(t)}]$, where $M_{-\omega_{\mathrm{obs}}}^{(t)}$ and $M_{\hat{\omega}_k}^{(t)}$ are rotational matrices due to the rotation of the observer and the object, respectively, and $v_{\mathrm{obs}}^{(t)}$ is the velocity of the observer (relative to its own reference frame at $t$). In this way, assuming objects move smoothly most of the time, if the observer's self motion information is known, the 3D location of each visible pixel can be predicted. If a pixel belongs to the background, $\omega_{K+1} = 0$ and $v_{K+1} = 0$ ($K + 1$ is the background's index). Given the predicted 3D location, the target coordinate $(i', j')_k^{(t+1)}$ of the pixel on the image and its new depth $D'_k(i, j)^{(t+1)}$ can be calculated. This prediction of pixel movement allows predicting the image $I'^{(t+1)}$ and depth $D'^{(t+1)}$ by weighting the colors and depth of all pixels predicted to land near each discrete pixel grid of frame $t + 1$.

We use the weights of bilinear interpolation to decide how much each source pixel at $t$ contributes to drawing the four pixel grids surrounding the predicted landing location of that pixel in frame $t + 1$, assuming the pixel fully belongs to one object or the background. Because each pixel may belong to any of the $K$ object and the background, it has $K + 1$ potential landing point at $t + 1$ each with a probability prescribed by the segmentation output for that pixel. Therefore, the contribution weight is further multiplied by the segmentation probability that attributes the source pixel to the object or the background which would bring the source pixel to the target location at $t + 1$ (more details in appendix). The sum of all contribution weights a target pixel receives indicates how much that pixel can be predicted by warping. When the sum is smaller than 1, the final prediction for the pixel will be a weighted average of the prediction by warping and by imagination.

### 2.4.2 IMAGINATION

For the pixels not fully predictable by warping current image, we mix the prediction based on warping with a prediction learned from the statistical regularity of scenes which we call "imagination". To do so, We learn a function $g$ that "imagines" the appearance $I'^{(t+1)}_{k\mathrm{Imag}} \in \mathbb{R}^{w \times h \times 3}$ and the pixel-

wise depth $\boldsymbol{D}'^{(t+1)}_{k\text{Imag}} \in \mathbb{R}^{w \times h}$ of each object $k$ or background in the next frame, and weight them with predicted probabilities that each pixel in the next frame belongs to each object or the background $\boldsymbol{\pi}'^{(t+1)}_{k\text{Imag}} \in \mathbb{R}^{w \times h}$. The function takes as input portion of the current image corresponding to each object $\boldsymbol{I}^{(t)} \odot \boldsymbol{\pi}^{(t)}_k$ and logarithm of the inferred depth $\log(\boldsymbol{D}^{(t)}) \odot \boldsymbol{\pi}^{(t)}_k$, both extracted by element-wise multiplying with the probabilistic segmentation mask $\boldsymbol{\pi}^{(t)}_k$, the information of the camera's self motion, and the inferred location and motion of that object:

$$\{\boldsymbol{I}'^{(t+1)}_{k\text{Imag}}, \boldsymbol{D}'^{(t+1)}_{k\text{Imag}}, \boldsymbol{\pi}'^{(t+1)}_{k\text{Imag}}\} = g(\boldsymbol{I}^{(t)}_i \odot \boldsymbol{\pi}^{(t)}_k, \log(\boldsymbol{D}^{(t)}) \odot \boldsymbol{\pi}^{(t)}_k, \hat{\boldsymbol{x}}^{(t)}_k, \hat{\boldsymbol{v}}^{(t)}_k, \hat{\omega}^{(t)}_k, \omega^{(t)}_{\text{obs}}) \qquad (2)$$

The "imagination" specific for each object and the background can then be merged using the weights prescribed by $\boldsymbol{\pi}'^{(t+1)}_{1:K\text{Imag}}$: $\boldsymbol{I}'^{(t+1)}_{\text{Imag}} = \sum_k \boldsymbol{I}'^{(t+1)}_{k\text{Imag}} \odot \boldsymbol{\pi}'^{(t+1)}_{k\text{Imag}}$, and $\boldsymbol{D}'^{(t+1)}_{\text{Imag}} = \sum_k \boldsymbol{D}'^{(t+1)}_{k\text{Imag}} \odot \boldsymbol{\pi}'^{(t+1)}_{k\text{Imag}}$. This is similar to the way several other OCRL methods (Burgess et al., 2019; Locatello et al., 2020) reconstruct the current image without using geometric knowledge, except that our "imagination" predicts the next image by additionally conditioning on the motion information of the observer and objects.

### 2.4.3 COMBINING WARPING AND IMAGINATION

The final predicted image or depth map are weighted average of the prediction made by warping the current image or inferred depth map and the corresponding predictions by Imagination network:

$$\boldsymbol{I}'^{(t+1)} = \boldsymbol{I}'^{(t+1)}_{\text{Warp}} \odot \boldsymbol{W}_{\text{Warp}} + \boldsymbol{I}'^{(t+1)}_{\text{Imag}} \odot (1 - \boldsymbol{W}_{\text{Warp}}) \qquad (3)$$

Here, each element of the weight map for the prediction based on warping $\boldsymbol{W}_{\text{Warp}} \in \mathbb{R}^{w \times h}$ is $W_{\text{Warp}}(p, q) = \max\{\sum_{k,i,j} w(i, j, p, q), 1\}$ and $w(i, j, p, q)$ is the contribution weight of pixel $(i, j)$ at $t$ for drawing pixel $(p, q)$ at $t + 1$ (more details in appendix). The intuition is that imagination is only needed when not enough pixels from $t$ will land near $(p, q)$ at $t + 1$ based on the predicted optical flow. The same weighting applies for generating the final predicted depth $\boldsymbol{D}'^{(t+1)}$.

### 2.4.4 LEARNING OBJECTIVE

Above, we have explained how to predict the spatial states of each object, $\boldsymbol{x}'^{(t+1)}_k$ and $p'^{(t+1)}_{\phi_k}$, the next image $\boldsymbol{I}'^{(t+1)}$ and depth map $\boldsymbol{D}'^{(t+1)}$ based on object-centric representation extracted by a function $f$ from the current and previous images $\boldsymbol{I}^{(t)}_i$ and $\boldsymbol{I}^{(t-1)}_i$, the depth $\boldsymbol{D}'^{(t)}$ extracted by a function $h$, combined with the prediction from object-based imagination function $g$ that are all to be learned. Among the three prediction targets, only the ground truth of visual input $\boldsymbol{I}^{(t+1)}$ is available, while the other can only be inferred by $f$ and $h$ from $\boldsymbol{I}^{(t+1)}$. Therefore, for the prediction targets other than $\boldsymbol{I}^{(t+1)}$, we use the self-consistent loss between the predicted value based on $t$ and $t - 1$ and the inferred value based on $t + 1$ as additional regularization terms to learn $f$ and $g$.

To learn the functions $f$, $g$ and $h$, we approximate them with deep neural networks with parameters $\theta$ and optimize $\theta$ to minimize the following loss function:

$$L = L_{\text{image}} + \lambda_{\text{depth}} L_{\text{depth}} + \lambda_{\text{spatial}} L_{\text{spatial}} + \lambda_{\text{map}} L_{\text{map}} \qquad (4)$$

Here, $L_{\text{image}} = \text{MSE}(I'^{(t+1)}, I^{(t+1)})$ is the image prediction error. $L_{\text{depth}} = \text{MSE}(\log(D'^{(t+1)}), \log(\hat{D}^{(t+1)}))$ is the error between the predicted and inferred depth. These provide major teaching signals. $L_{\text{spatial}} = \sum_{k=1}^{K} |\sum_{l=1}^{K+1} r_{kl} \boldsymbol{x}'^{(t+1)}_{kl} - \hat{\boldsymbol{x}}^{(t+1)}_k|^2 - \sum_{k=1}^{K} \min\{|\hat{\boldsymbol{x}}^{(t)}_{\text{rand}} - \hat{\boldsymbol{x}}^{(t)}_k|, \delta\} + \sum_{k=1}^{K} |\hat{\boldsymbol{x}}^{(t)}_k - \sum_{i,j} \hat{\boldsymbol{m}}^{(t)}_{i,j} \pi_{kij}|^2 + \sum_{k=1}^{K} D_{\text{KL}}(\hat{\boldsymbol{p}}^{(t+1)}_{\phi_k} || \sum_{l=1}^{K+1} r_{kl} \boldsymbol{p}'^{(t+1)}_{\phi_l})$ is the self-consistent loss on spatial information prediction. The first term is the error between inferred and predicted location of each object, while the calculation of the predicted location incorporates soft matching between objects in consecutive frames ($r_{kl}$ is the matching probability between any objects in two frames). The second term is the negative term of contrastive loss, which we found empirically helpful to prevent the network from reaching a local minimum where all objects are inferred at the same location relative to the camera (and covering minimal regions of the picture). $\hat{\boldsymbol{x}}_{\text{rand}}$ is the inferred object location from a random sample within the same batch. The third term penalizes the discrepancy between the inferred object location and the average location of pixels in

its segmentation mask. The last term is the KL-divergence between the predicted and inferred pose for each object at $t + 1$. $L_{\text{map}} = \text{ReLu}(10^{-4} - \boldsymbol{\pi}_{1:K}) + \sum_{k=1}^{K} \text{ReLu}(\bar{\boldsymbol{\pi}}_k - 0.1) + \boldsymbol{\pi}_k \cdot \boldsymbol{\pi}_l$, for $k \neq l$ avoids loss of gradient due to zero probability of object belonging and discourages overlap between maps of different objects.

## 2.5 DATASET

We procedurally generated a dataset composed of 306K triplets of images captured by a virtual camera with a field of view of 90 degrees in a square room. The camera translates horizontally and pans with random small steps between consecutive frames to facilitate the learning of depth perception. 3 objects with random shape, size, surface color or textures are spawned at random locations in the room and each moves with a randomly selected constant velocity and panning speed. The translation and panning of the the camera relative to its own reference frame and its intrinsic are known to the networks. No other ground truth information is provided. The first two frames serve as data and the last frame serves as the prediction target at $t + 1$. An important difference between this dataset and other commonly used synthetic datasets for OCRL, such as CLEVRset (Johnson et al., 2017) and GQN data (Eslami et al., 2018) is that more complex and diverse textures are used on both the objects and the background. We further evaluated on a richer version of the Traffic dataset (Henderson & Lampert, 2020) in A.4.

## 2.6 COMPARISON WITH OTHER WORKS

To compare our work with the state-of-the-art models of unsupervised object-centric representation learning, we trained MONet (Burgess et al., 2019), slot-attention (Locatello et al., 2020) and GEN-ESIS[2] (Engelcke et al., 2021) on the same dataset. Although these models are trained on single images, all images of each triplets are used for training.

To address the concern that the original configurations of the models are not optimized for more difficult dataset, we additionally trained variants of some of the models with large network size. For MONet, we tested channel numbers of [32, 64, 128, 128] (MONet-128) and [32, 64, 128, 256, 256] (MONet-128-bigger) for the hidden layers of encoder of the component VAE (while the original paper used [32, 32, 64, 64]) and adjusted decoder layers' sizes accordingly, and increased the base channel from 64 to 128 for its attention network. For slot attention, we tested a variant which increased the number of features in the attention component from 64 to 128 (slot-attention-128). Slot numbers were chosen as 4 except for GENESIS.

| MODEL | ARI-FG | IOU |
|---|---|---|
| MONET | 0.31 | 0.08 |
| MONET-128 | 0.33 | 0.22 |
| MONET-128-BIGGER | 0.36 | 0.20 |
| SLOT-ATTENTION | 0.40 | 0.34 |
| SLOT-ATTENTION-128 | 0.34 | **0.38** |
| OUR MODEL (OPPLE) | **0.46** | 0.35 |

Table 1: Performance of different models on object segmentation.

## 3 RESULTS

After training the networks, we evaluate them on 4000 test images unused during training but generated randomly with the same procedure, thus coming from the same distribution. We compare the performance of different models mainly on their segmentation performance. Additionally, we demonstrate the ability unique to our model: inferring locations of objects in 3D space and the depth of the scene. The performance of depth perception is illustrated in the appendix.

## 3.1 OBJECT SEGMENTATION

Following prior works (Greff et al., 2019; Engelcke et al., 2019; 2021), we evaluated segmentation with the Adjusted Rand Index of foreground objects (ARI). In addition, for each image, we matched

---

[2]We failed to obtain reasonable result by training GENESIS V2 on our dataset, thus we adopted a GENESIS network pre-trained on GQN dataset and retrained on our dataset with K=7. The result is included in the appendix as we are not confident that we have exhausted hyper-parameter tuning for this model

ground-true objects and background with each of the segmented class by ranking their Intersection over Union (IoU) and quantified the average IoU over all foreground objects. The performance is summarized in table 2.6 .

Our model outperforms all compared models on ARI and is second to a slot-attention-128 in IoU. As shown in Figure 2, MONet appear to heavily rely on color to group pixels into the same masks. Even though some of these models almost fully designate pixels of an object to a mask, the masks lack specificity in that they often include pixels with similar colors from other objects or background. Patterns on the backgrounds are often treated as objects as well.

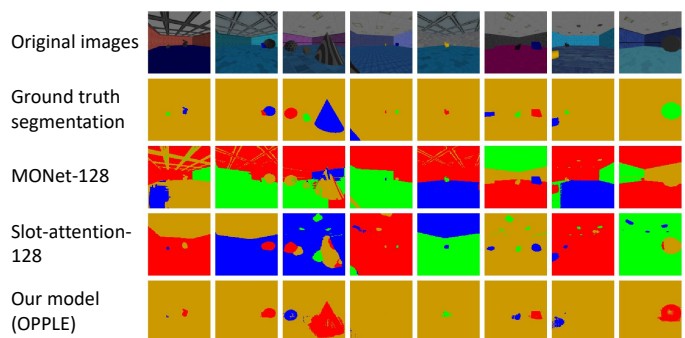

Figure 2: Example of object segmentation by different models

We postulate there may be fundamental limitation in the approach that learns purely from static discrete images. Patches in the background with coherent color offer room to compress information similarly as objects with coherent colors do, and their shapes re-occur across images just as other objects. Our model is able to learn object-specific masks because these masks are used to predict optical flow specific to each object. A wrong segmentation would generate large prediction error even if the motion of an object is estimated correctly. Such prediction error forces the masks to concentrate on object surface. They emerge first at object boundaries where the prediction error is the largest and gradually grow inwards during training. Figure 3A-C further compares the distribution of IoU across models. Only OPPLE and slot-attention-128 show bi-modal distributions while MONet is skewed towards 0. Figure 3G plots each object's distance and size on the picture with colors corresponding to their IoUs in our model. Objects with poor segmentation (blue dots) are mostly far away from the camera and occupy few pixels. This is because motion of farther objects causes less shift on the images and thus provides weaker teaching signal. For other models, blue dots are more spread even for near objects (See appendix).

## 3.2   OBJECT LOCALIZATION

The Object Extraction Network infers object location relative to the camera. We convert the inferred locations to angles and distance in polar coordinate relative to the camera. Figure 3E-F plot the true and inferred angles and distance, color coded by objects' IoUs. For objects well segmented (red dots), their angles are estimated with high accuracy (concentrated on the diagonal in E). Distance estimation is negatively biased for farther objects, potentially because the regularization term on the distance between the predicted and inferred object location at frame $t+1$ favors shorter distance when estimation is noisy. Note that the ability to explicitly infer object's location is not available in other models compared.

## 3.3   MEANINGFUL LATENT CODE

Because a subset of the latent code (10 dimensions) was used to calculate object matching scores between frames in order to soft-match objects, this should force the object embedding $z$ to be similar for the same objects. We explored the geometry of the latent code by examining whether the nearest neighbours of each of the object in the test data with IoU $> 0.5$ are more likely to have the same property as themselves. 772 out of 3244 objects' nearest neighbour had the same shape (out of 11 shapes) and 660 objects' nearest neighbour had the same color or texture (out of 15). These numbers are 28 to 29 times the standard deviation away from the means of the distribution expected if the nearest neighbour were random (Figure 3H). This suggests the latent code reflects meaningful features of objects. However, texture and shape are not the only factors determining latent code, as we found the variance of code of all objects with the same shape and texture to still be big.

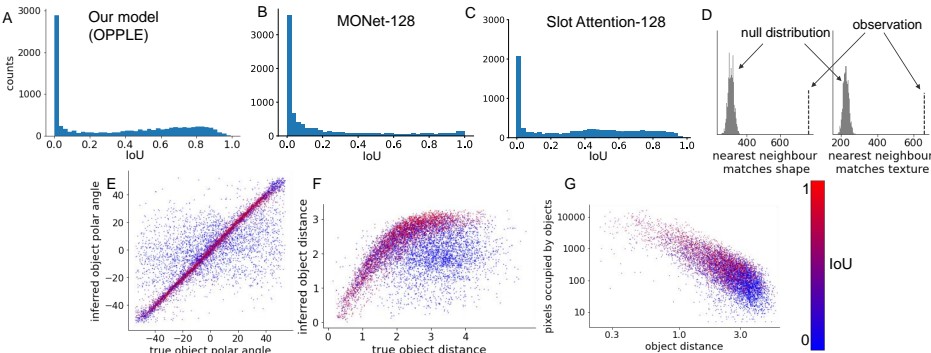

Figure 3: **A-C**: distribution of IoU. All models have IoU $< 0.01$ for about 1/4 of objects. Only OPPLE shows a bi-modal distribution while other models' IoU are more skewed towards 0. **D** The numbers of objects sharing the same shape or texture with their nearest neighbour objects in latent space are significantly above chance. **E-F**: object localization accuracy of OPPLE for object's polar angle and distance relative to the camera. Each dot is a valid object with color representing its segmentation IoU. Angle estimation is highly accurate for well segmented objects (red dots). Distance is under-estimated for farther objects. **G**: objects with failed segmentation (blue dots) are mostly far away and occupying few pixels.

## 4 RELATED WORK

Our work is on the same tracks as two recent trends in machine learning: object-centric representation (Locatello et al., 2020) and self-supervised learning (Chen et al., 2020). We follow the same logic as self-supervised learning: learning to predict part of the data based on another part forces a neural network to learn important structures in the data. However, most of the existing works in self-supervised learning do not focus on object-based representation, but instead encode the entire scene as one vector. Other works on object-centric representations overcome this by assigning one representation to each object, as we do. Although works such as MONet (Burgess et al., 2019), IODINE (Greff et al., 2019), slot-attention (Locatello et al., 2020), GENESIS (Engelcke et al., 2019) and PSGNet (Bear et al., 2020) can also segment objects and some of them can "imagine" complete objects based on codes extracted from occluded objects or draw objects in the correct order consistent with occlusion, few works explicitly infer an object's location in 3D space together with segmentation purely by self-supervised learning, with the exception of a closely related work O3V (Henderson & Lampert, 2020). Both our works learn from videos. One major distinction is that O3V interleaves spatial and temporal convolution, thus it still require video as input at test time. In contrast, our three major networks process each image independently. Therefore, once trained, our network can generalize to single images. Another distinction from Henderson & Lampert (2020) and many other works is that our model learns from prediction instead of reconstruction. Contrastive-learning of a structured world model (Kipf et al., 2019) also learns object masks and predicts their future states by linking each object mask with a node in a Graphic Neural Network (GNN). The order of mapping object slot to nodes of GNN is fixed through time. This arrangement may become infeasible with combinatorial number of possible objects as the order of assigning different objects to a limited number of nodes may not be consistent across scenes. We solve this by a soft matching of object representation between different time points. On the neuroscience side, our work is highly motivated by recent works on predictive learning (O'Reilly et al., 2021) which also yields view-invariance representation while self-motion signal is available. O'Reilly et al. (2021) used biologically plausible but less efficient learning and applied their model to an easier dataset with objects without background, and did not learn object localization. We should note that explicit spatial localization and depth perception were not pursued in previous works on self-supervised object-centric learning, and the images in our dataset have significantly richer texture information than those demonstrated in previous works (Burgess et al., 2019; Kipf et al., 2019), making the task more challenging. SAVi++(Elsayed et al., 2022) is closely related to our work but as it requires external information of optical flow, we cannot make a fair comparison here. Although view synthesis is not our central goal, the principle illustrated here can be combined with recent advancement in 3D-aware image synthesis (Wiles et al., 2020).

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

# A  APPENDIX

## A.1  PSEUDO CODE OF THE OPPLE FRAMEWORK

---

**Algorithm 1** Developing object-centric representation by predicting future scene

---

**Input:** images $\boldsymbol{I}^{(t-1)}, \boldsymbol{I}^{(t)} \in \mathbb{R}^{w \times h \times 3}$, self-motion $\boldsymbol{v}_{\mathrm{obs}}^{(t-1)}, \omega_{\mathrm{obs}}^{(t-1)}, \boldsymbol{v}_{\mathrm{obs}}^{(t)}, \omega_{\mathrm{obs}}^{(t)}$
**Output:** prediction $\boldsymbol{I}'^{(t+1)}$, segmentation $\boldsymbol{\pi}_{1:K+1}^{(t-1)}, \boldsymbol{\pi}_{1:K+1}^{(t)}$, objects' codes $\boldsymbol{z}_{1:K}^{(t-1)}, \boldsymbol{z}_{1:K}^{(t)}$, objects'
    locations and poses $\hat{\boldsymbol{x}}_{1:K}^{(t-1)}, \boldsymbol{p}_{\phi_{1:K}}^{(t-1)}, \hat{\boldsymbol{x}}_{1:K}^{(t)}, \boldsymbol{p}_{\phi_{1:K}}^{(t)}$
    **for** $\tau = \{t-1, t\}$ **do**
        scene code $e^{(\tau)} \leftarrow$ U-NetEncoder$_{f_\theta}(\boldsymbol{I}^{(\tau)})$
        object code $\boldsymbol{z}_{1:K}^{(\tau)}$, location $\hat{\boldsymbol{x}}_{1:K}^{(\tau)}$, pose $\boldsymbol{p}_{\phi_{1:K}}^{(\tau)} \leftarrow$ LSTM$_{f_\theta}(e^{(\tau)})$
        background code $\boldsymbol{z}_{K+1} = 0$
        depth $\boldsymbol{D}^{(\tau)} \leftarrow h_\theta(\boldsymbol{I}^{(\tau)})$
        segmentation mask $\boldsymbol{\pi}_{1:K+1}^{(\tau)} \leftarrow$ Softmax (U-NetDecoder$_{f_\theta}(\boldsymbol{I}^{(\tau)}, \boldsymbol{z}_{1:K}^{(\tau)}), 0)$
    **end for**
    object matching scores $r_{kl} \leftarrow$ RBF$(\boldsymbol{z}_k^{(t)}, \boldsymbol{z}_l^{(t-1)}), k, l \in 1 : K+1$
    **for** $k \leftarrow 1$ to $K$ **do**
        object motion $\hat{\boldsymbol{v}}_{1:K}, \boldsymbol{\omega}_{1:K} \leftarrow r_{k,l}, \hat{\boldsymbol{x}}_k^{(t)}, \hat{\boldsymbol{x}}_l^{(t-1)}, \boldsymbol{p}_{\phi_k}^{(t)}, \boldsymbol{p}_{\phi_l}^{(t-1)}, l = 1 : K+1$
        onject-specific optical flow$_k \leftarrow \hat{\boldsymbol{v}}_{1:K}, \boldsymbol{\omega}_{1:K}, \boldsymbol{v}_{\mathrm{obs}}, \omega_{\mathrm{obs}}, \boldsymbol{D}^{(t)}$
    **end for**
    $\boldsymbol{I}'^{(t+1)}_{\mathrm{warp}} \leftarrow$ Warp$(\boldsymbol{I}^{(t)}, \text{optical flow}_{1:K+1})$
    $\boldsymbol{I}'^{(t+1)}_{\mathrm{imagine}} \leftarrow g_\theta(\boldsymbol{I}^{(t)} \odot \boldsymbol{\pi}_{1:K+1}^{(t)}, \log(\boldsymbol{D}^{(t)}) \odot \boldsymbol{\pi}_{1:K+1}^{(t)}, \boldsymbol{v}_{\mathrm{obs}}, \omega_{\mathrm{obs}}, \hat{\boldsymbol{v}}_{1:K+1}, \hat{\boldsymbol{x}}_{1:K})$
    final image prediction: $\boldsymbol{I}'^{(t+1)} \leftarrow \boldsymbol{I}'^{(t+1)}_{\mathrm{warp}}, \boldsymbol{I}'^{(t+1)}_{\mathrm{imagine}}$, warping weights
    update parameters: $\theta \leftarrow \theta - \gamma \nabla_\theta[|\boldsymbol{I}'^{(t+1)} - \boldsymbol{I}^{(t+1)}|^2 + \text{regularization loss}]$

---

## A.2  NETWORK TRAINING AND DATASET

We trained the three networks jointly using ADAM optimization Kingma & Ba (2014) with a learning rate of $3e-4$, $\epsilon = 1e-6$ and other default setting in PyTorch, with a batch size of 24. 40 epochs were trained on the dataset. We set $\lambda_{\mathrm{spatial}} = 1.0$, $\lambda_{\mathrm{depth}} = 0.1$ and $L_{\mathrm{map}} = 0.005$.

Images in the dataset are rendered in a custom Unity environment at a resolution of $512 \times 512$ and downsampled to $128 \times 128$ resolution for training. Images are rendered in sequence of 7 time steps for each scene. In each scene, a room with newly selected textures and objects is created, then the objects and camera are placed randomly in the room and checked for possible colliding trajectories, once a configuration of locations and movement direction of objects and camera are found where they don't collide, we start recording the scene. We do this pre-check to reduce the redundancy of getting bad datasets in our recordings. Camera moves with random steps and sways with random rotations between consecutive frames. All possible sequential triplets out of the recorded 7 frames (including frames equally spaced by 0, 1, and 2 frames) form our training samples. For our dataset we use 200,000 scenes which give us 2,400,000 datapoints.

The model was implemented in PyTorch and trained on NVidia RTX 6000. We will release the code and dataset upon publication of the manuscript.

## A.3  PERFORMANCE ON DEPTH PERCEPTION

We demonstrate a few example images and the inferred depth. Our network can capture the global 3D structure of the scene, although details on object surfaces are still missing. Because background occurs in every training sample, the network appears to bias the depth estimation on objects towards the depth of the walls behind, as is also shown in the scatter plot.

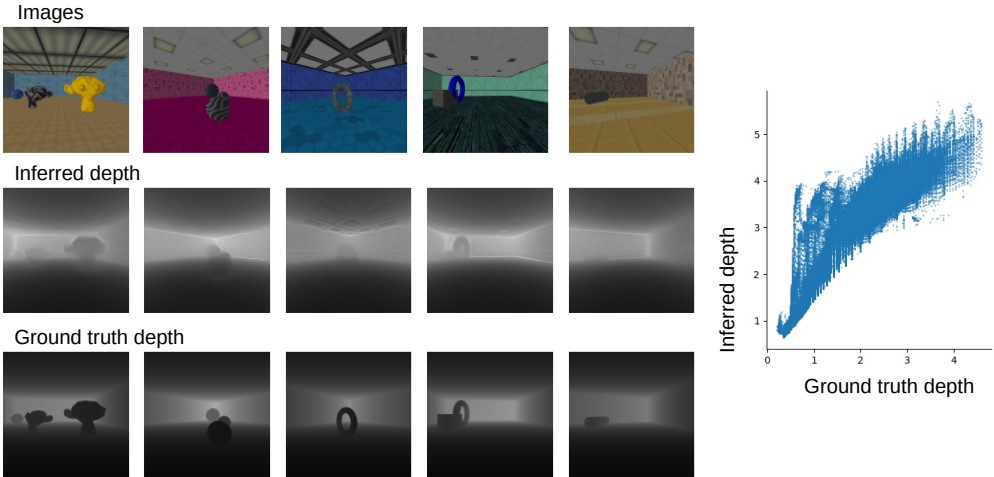

Figure 4: Comparison between ground truth depth and inferred depth

## A.4 Segmentation performance on more challenging images

We further evaluated our model on images of running vehicles in virtual towns generated by Carla environment (Dosovitskiy et al., 2017), following the approach of (Henderson & Lampert, 2020) but with much more diverse scenes (the traffic dataset used in (Henderson & Lampert, 2020) were collected in one street with camera viewing from one side of the vehicle while we trained and tested our model across different cities with all possible viewing angles). This dataset is much more challenging as it includes more realistic lighting condition (e.g., mirror reflection) and much more complex scene structure. As shown, across different viewing angles, the model can always segment the major car in view correctly, although with some artifact in segmentation mask. One difficult situation is when the road's color is very homogeneous, because the cars mostly move along the road, treating part of the road as object does not increase prediction error by warping and the model turn to make such mistake. Because some cars distant from the camera are very small and barely detectable, we calculate IoU weighted by the size of object, this yields a size-weighted IoU of 0.44.

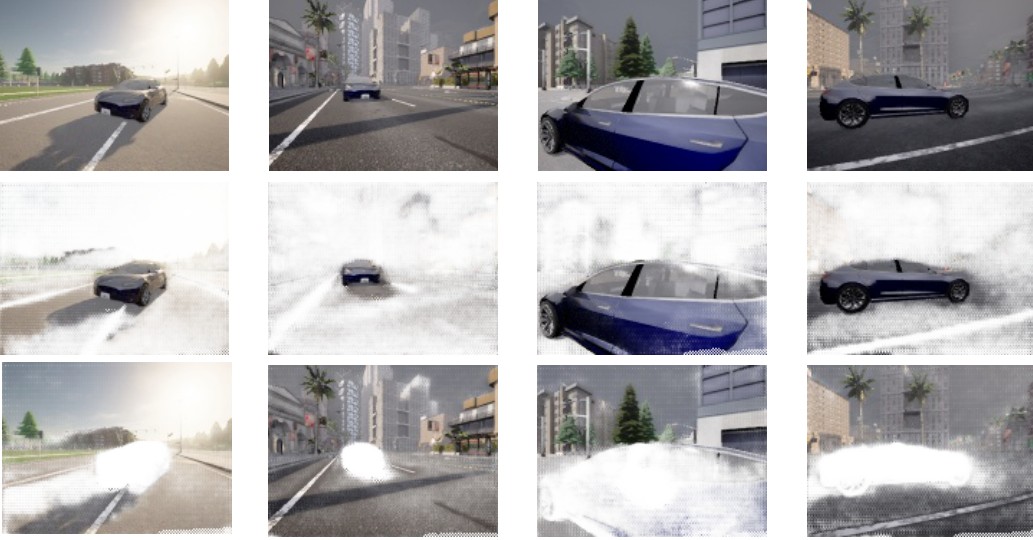

Figure 5: Segmentation performance on images of cars in virtual towns generated by Carla. Top: input images, middle: image weighted by the inferred mask for the object slot containing the car, bottom: image weighted by the inferred mask for the background. Other slots (3 in total in training) are neglected.

## A.5 METHOD

### A.5.1 PREDICTION BY WARPING

We first describe the prediction of part of the next image by warping the current image. Here we consider only rigid objects and the fates of all visible pixels belonging to an object. With depth $\boldsymbol{D}^{(t)} \in \mathbb{R}^{w \times h} = h_\theta(\boldsymbol{I}^{(t)})$ of all pixels in a view inferred by the Depth Perception network based on visual features in the image $\boldsymbol{I}^{(t)}$, the 3D location of a pixel at any coordinate $(i,j)$ in the image, where $|i| \leq \frac{w-1}{2}, |j| \leq \frac{h-1}{2}$, can be determined given the focal length $d$ of the camera as $\hat{\boldsymbol{m}}_{(i,j)}^{(t)} = \frac{D^{(t)}(i,j)}{\sqrt{i^2+j^2+d^2}} \cdot [i, d, j]$. Here, we take the coordinate of the center of an image as (0,0). On the other hand, with the inferred $\hat{\boldsymbol{x}}_k^{(t)}$ and $\hat{\boldsymbol{x}}_k^{(t-1)}$, the current and previous locations of the object $k$ that the pixel $(i,j)$ belongs to, from $\boldsymbol{I}^{(t)}$ and $\boldsymbol{I}^{(t-1)}$ respectively, we can estimate the instantaneous velocity of the object $\hat{\boldsymbol{v}}_k^{(t)} = \hat{\boldsymbol{x}}_k^{(t)} - \hat{\boldsymbol{x}}_k^{(t-1)}$. Similarly, with the inferred the current and previous pose probabilities of the object, $\boldsymbol{p}_{\phi_k}^{(t)}$ and $\boldsymbol{p}_{\phi_k}^{(t-1)}$, we can obtain the likelihood of its angular velocity

$$p(\phi_k^{(t)}, \phi_k^{(t-1)} \mid \omega_k^{(t)} = \omega) \propto \sum_{\substack{\gamma_1, \gamma_2, \gamma_1 - \gamma_2 \in \\ \{\omega - 2\pi, \omega, \omega + 2\pi\}}} p(\phi_k^{(t)} = \gamma_1) \cdot p(\phi_k^{(t-1)} = \gamma_2) \tag{5}$$

.

By additionally imposing a prior distribution (we use Von Mises distribution) over $\omega_k^{(t)}$ that favors slow rotation, we can obtain the posterior distribution of the object's angular velocity $p(\omega_k^{(t)} \mid \phi_k^{(t)}, \phi_k^{(t-1)})$, and eventually the posterior distribution of the object's next pose $p(\phi_k^{(t+1)} \mid \phi_k^{(t)}, \phi_k^{(t-1)})$.

Assuming a pixel $(i,j)$ belongs to object $k$, using the estimated motion information $\hat{\boldsymbol{v}}_k^{(t)}$ and $p(\omega_k^{(t)} \mid \phi_k^{(t)}, \phi_k^{(t-1)})$ of the object, together with the current location and pose of the object and the current 3D location $\hat{\boldsymbol{m}}_{(i,j)}^{(t)}$ of the pixel, we can predict the 3D location $\boldsymbol{m'}_{k,(i,j)}^{(t+1)}$ of the pixel at the next moment as

$$\boldsymbol{m'}_{k,(i,j)}^{(t+1)} = \boldsymbol{M}_{-\omega_{\text{obs}}}^{(t)} [\boldsymbol{M}_{\hat{\omega}_k}^{(t)}(\hat{\boldsymbol{m}}_{(i,j)}^{(t)} - \hat{\boldsymbol{x}}_k^{(t)}) + \hat{\boldsymbol{x}}_k^{(t)} + \hat{\boldsymbol{v}}_k^{(t)} - \boldsymbol{v}_{\text{obs}}^{(t)}] \tag{6}$$

where $\boldsymbol{M}_{-\omega_{\text{obs}}}^{(t)}$ and $\boldsymbol{M}_{\hat{\omega}_k}^{(t)}$ are rotational matrices due to the rotation of the observer and the object, respectively, and $\boldsymbol{v}_{\text{obs}}^{(t)}$ is the velocity of the observer (relative to its own reference frame at $t$). In this way, assuming objects move smoothly most of the time, if the self motion information is known, the 3D location of each visible pixel can be predicted. If a pixel belongs to the background, $\omega_{K+1} = 0$ and $\boldsymbol{v}_{K+1} = 0$ ($K+1$ is the background's index). Given the predicted 3D location, the target coordinate $(i', j')_k^{(t+1)}$ of the pixel on the image and its new depth $D'_k(i,j)^{(t+1)}$ can be calculated. This prediction of pixel movement allows predicting the image $\boldsymbol{I'}^{(t+1)}$ and depth $\boldsymbol{D'}^{(t+1)}$ by weighting the colors and depth of pixels predicted to land near each pixel at the discrete grid of the next frame, as explained in Sec 2.4.2.

### A.5.2 WARPING CONTRIBUTION WEIGHT

As the object attribution of each pixel is not known but is inferred by $f_{\text{obj}}(\boldsymbol{I}^{(t)})$, it is represented for every pixel as a probability of belonging to each object and the background $\boldsymbol{\pi}_k^{(t)}$, $k = 1, 2, \cdots, K+1$. Therefore, the predicted motion of each pixel should be described as a probability distribution over $K+1$ discrete target locations $p(\boldsymbol{x'}_{(i,j)}^{(t+1)}) = \sum_{k=1}^{K+1} \pi_{kij}^{(t)} \cdot \delta(\boldsymbol{x'}_{k,(i,j)}^{(t+1)})$, i.e., pixel $(i,j)$ has a probability of $\pi_{kij}^{(t)}$ to move to location $\boldsymbol{x'}_{k,(i,j)}^{(t+1)}$ at the next time point, for $k = 1, 2, \cdots, K+1$. With such probabilistic prediction of pixel movement for all visible pixel $(i,j)^{(t)}$, we can partially predict the colors of the next image at the pixel grids where some original pixels from the current view will land nearby by weighting their contribution:

$$\boldsymbol{I'}_{\text{Warp}}^{(t+1)}(p, q) = \begin{cases} \frac{\sum_{k,i,j} w_k(i,j,p,q) I^{(t)}(i,j)}{\sum_{k,i,j} w_k(i,j,p,q)}, & \text{if } \sum_{k,i,j} w_k(i,j,p,q) > 0 \\ 0, & \text{otherwise} \end{cases} \tag{7}$$

We define the weight of the contribution from any source pixel $(i,j)$ to a target pixel $(p,q)$ as

$$w_k(i,j,p,q) = \pi_{kij}^{(t)} \cdot e^{-\beta \cdot D'^{(t+1)}_k(i,j)} \cdot \max\{1 - |i'^{(t+1)}_k - p|, 0\} \cdot \max\{1 - |j'^{(t+1)}_k - q|, 0\} \quad (8)$$

The first term incorporates the uncertainty of which object a pixel belongs to. The second term $e^{-\beta \cdot D'^{(t+1)}_k(i,j)}$ resolves the issue of occlusion when multiple pixels are predicted to move close to the same pixel grid by down-weighting the pixels predicted to land farther from the camera. These last two terms mean that only the source pixels predicted to land within a square of of $2 \times 2$ pixels centered at any target location $(p,q)$ will contribute to the color $I'^{(t+1)}_{\text{Warp}}(p,q)$. The depth map $\boldsymbol{D}'^{(t+1)}_{\text{Warp}}$ can be predicted by the same weighting scheme after replacing $I^{(t)}(i,j)$ with each predicted depth $D'^{(t+1)}_k(i,j)$ assuming the pixel belongs to object $k$.

## A.6 DEPENDENCY OF SEGMENTATION PERFORMANCE ON OBJECT SIZE AND DISTANCE ACROSS MODELS

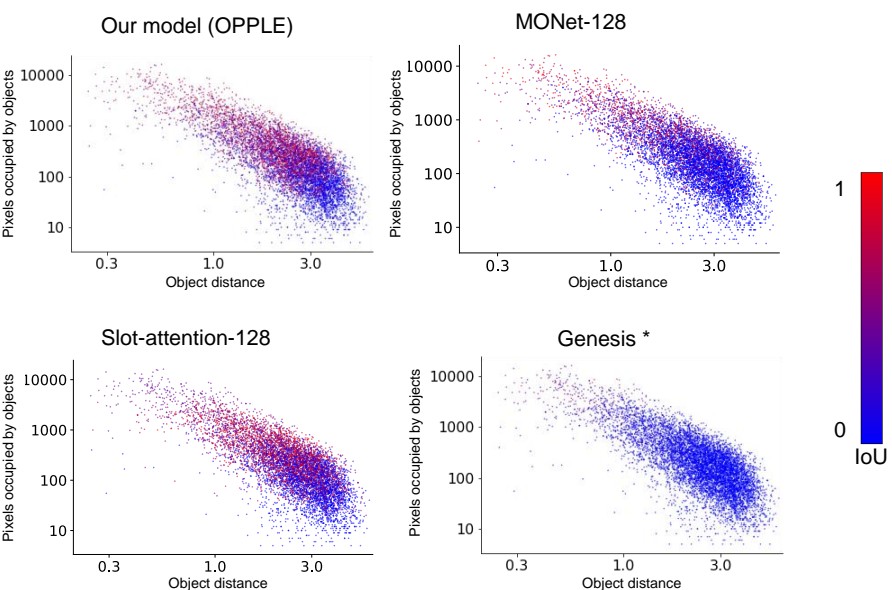

Figure 6: Comparison of segmentation quality with respect to the pixel occupancy and distance of the object from the observer.

We observe that for objects further away from the observer which occupies less pixels (because of the 2D perspective projection), the IOU goes down with the increase in the distance. We see this for all the compared models. This points to a requirement for robust segmentation to balance between the scales of the object. However, we are led to believe that even using an hierarchical structures to model this would fail as collating them together to the same scale would still be problematic, needing to weigh which scale is more important than others. Adding bigger motion to the object can help adding extra information for such small objects so that they are not missed during segmentation.

## A.7 COMPARISON WITH THE GENESIS MODEL

Genesis Engelcke et al. (2019) and the Genesis V2 Engelcke et al. (2021) models as mentioned in the main text are shown to perform very well in their manuscript. However, we were unable to train Genesis V2 with our dataset as we were consistently getting poor results. We decided to reuse the Genesis model pre-trained on the GQN dataset, then fine tuned it on our dataset. We show the segmentation result using this training procedure below. We keep this comparison in the appendix as we are not very confident how much longer it needs to be trained and how much more hyperparameter tuning it would need to improve. The comparison results are shown in the figure:7.

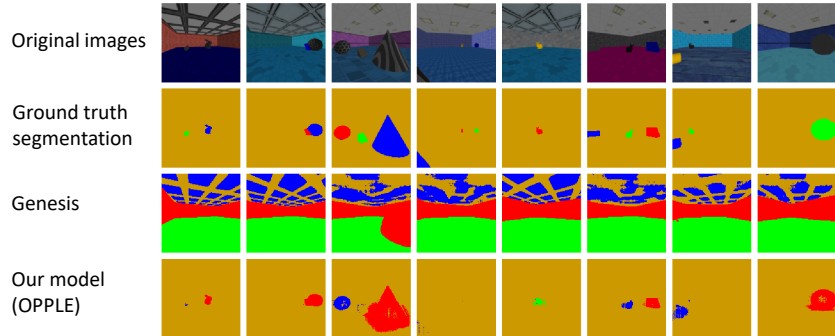

Figure 7: Example of object segmentation by different models

## A.8 ILLUSTRATION OF PREDICTION QUALITY

In the figure below, we display the first, second image, one of the masked objects, the predicted third image, and the ground truth of the third image, from top to the bottom. We provide reference lines and some circles to aid the comparison between images and evaluate the warping quality. Imagination quality can be inspected usually at one side of the predicted images.

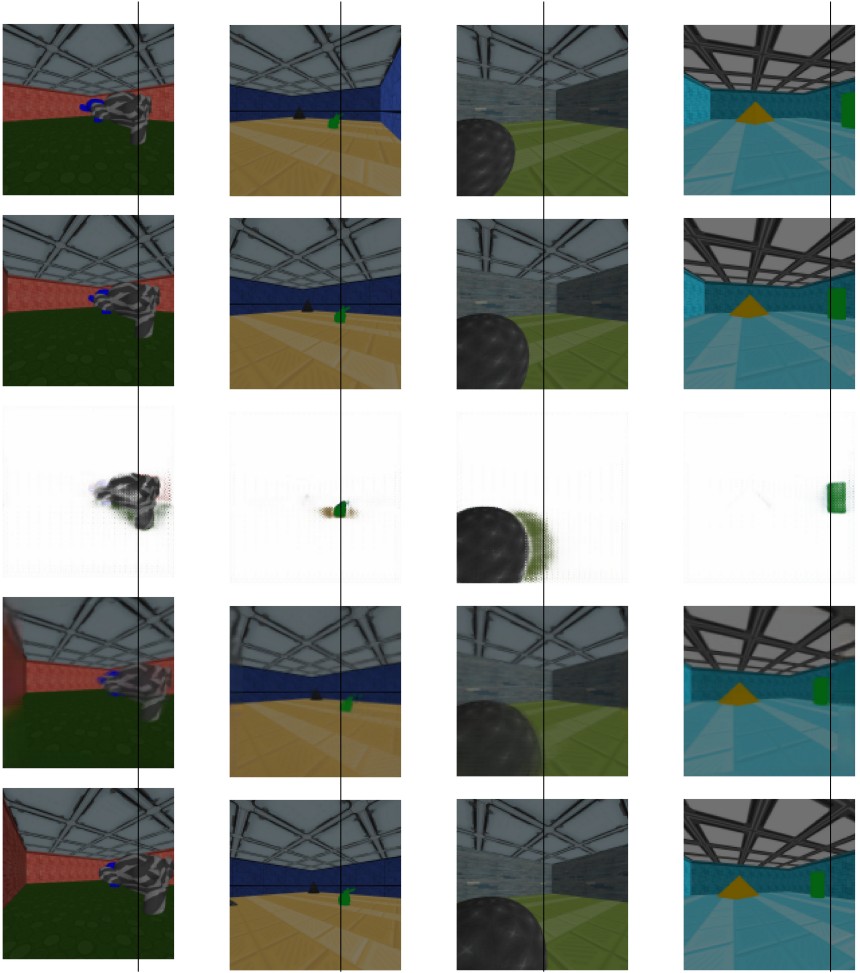

