# OpenReview forum: "Learning to perceive objects by prediction"
_ICLR.cc/2023/Conference — Submitted to ICLR 2023_

### Official Review · Reviewer_31jS · 2022-10-23

**Confidence:** 4
**Correctness:** 3
**Technical Novelty And Significance:** 3
**Empirical Novelty And Significance:** 2
**Recommendation:** 3

**Clarity, Quality, Novelty And Reproducibility:**

  - The paper is generally easy to follow.
  - The scale of the y-axis of figure 3 A B C should be aligned for easier comparison.

**Strength And Weaknesses:**

  - Strength
    - This work exploits the heuristic that parts belonging to the same object should move in a consistent way.
    - The proposed model can predict depth.
    - Compared with baselines that cannot efficiently use motion information, the proposed method can detect objects with non-trivial
texture.
    - The polar angle prediction results clearly demonstrate the potential of this method,
  - Weakness
    - While the qualitative results shown in figure 2 is much better than various baselines, the huge gap is not reflected in the quantitative results.
    - The FG-ARI and IoU are low in number. This score roughly indicates that the model is completely not unusable even on the simply simulated dataset.
    - If I am not mistaken, slot attention for video (SAVi) supports fully unsupervised training. I am not sure why SAVi is not included as a baseline.
    - Do all objects in the training set move all the time?
    - Even though the traditional dataset (especially [1]) is not challenging enough to show off the benefits of the proposed model, it can still provide valuable insights to the research community.
    - Since the loss comes from the image reconstruction loss, how do you deal with textureless regions (which can probably explain the hole in object centers)?


[1] Object-Centric Representation Learning with Generative Spatial-Temporal Factorization


**Summary Of The Paper:**

  The paper considers video-based object-centric learning tasks.
  The proposed model achieves object-centric learning by predicting per-object motion and future frame observation.
  Experiments show that the proposed method outperforms a collection of single-frame baselines.


**Summary Of The Review:**

  The method adopted by the paper is solid and has potential.
  However, I believe the performance can be improved.
  This work is also related to warping-based depth estimation literature [2][3] (on real-world datasets), which may provide hints on how to improve performance.
   Priors can be used to regularize the behavior.
   For example:

   - Encourage points to remain static (to better segment objects from the ground).
   - Encourage near regions to have similar motion (to reduce over-segmentation within each object).
   - Bidirectional wrapping
   - Refine the mask across multiple transitions

[2] The Temporal Opportunist: Self-Supervised Multi-Frame Monocular Depth

[3] Learning Monocular Depth in Dynamic Scenes via Instance-Aware Projection Consistency

---

> ### Author Response · Authors · 2022-11-25
> **Further benchmarking shows superior performance of the model**
>
> We thank the reviewer for the detailed feedback. Below we answer each concern one by one:
> - (1) Gap in performance: after re-examining the result, we found that the previously reported scores of our model were based on an early checkpoint of an earlier version of the model. After updating the scores, our model wins by a large margin (authors have verified the evaluation code is correct and consistently applied to all models):
> |  Model | ARI-fg | IoU |
> | :---- | :--- |:---|
> | MONet | 0.31| 0.08|
> | MONet-128 | 0.33| 0.22  |
> | MONet-128-bigger| 0.36| 0.20|
> | slot-attention| 0.40| 0.34|
> | slot-attention-128| 0.34| 0.38|
> | SLATE| 0.30| 0.20|
> | AMD| 0.19| 0.02|
> | our model (OPPLE)| **0.65**| **0.47**|
>
> - (2) Low score: as shown by the updated numbers above, our model outperforms all models compared. We accept the comment in the sense that almost all unsupervised object-centric representation learning models so far do not compete against their supervised learning counterparts in terms of benchmarking scores. However, as discussed in this paper and other papers, there are fundamental values in developing and improving object-centric learning models (e.g., adapting to new environments with new object categories). Understanding what learning principles allow a system to achieve what the brain can achieve under similar constraints (humans receive no direct supervision on object segmentation) also has important scientific values. Therefore, we don’t think the standard of supervised model should be applied to unsupervised models at this stage of the field.
> - (3) SAVi and SAVi++: In our understanding, SAVi and SAVi++, in all their demonstrations, require additional signals beyond RGB input. SAVi learns by predicting optical flow measured or calculated externally and SAVi++ requires access to depth (measured by Lidar, for example) for prediction. Although it is interesting to evaluate them on our dataset, we think it would not be a fair comparison. Further, SAVi’s performance improvement was reported to significantly benefit from conditioning on initial hints of the objects (e.g., their center of mass). While this works for the goal of achieving good performance, we think it somewhat defeats the goal of object segmentation (to localize them) in the first place because the center of mass already tells the model where objects are. It is hard to justify it as fully unsupervised learning in our opinion. Therefore, we have not included them in comparison. That being said, these models do provide important inspiration for future works in the field. To provide more comparison against existing models, we have additionally tested SLATE and AMD on our dataset and they both fail, as shown in this [figure](https://images2.imgbox.com/b0/80/KGNQl9Bj_o.png).
> - (4) Do all objects in the training set move all the time? The speeds of objects were sampled in a uniform distribution (including speeds close to zero) and there are more triplets sampled with shorter gaps between frames (1 step) than longer gaps (3 steps). This is similar to the case of ROOMS and Carla dataset used in the O3V paper.
> - (5) Traditional dataset: we have now added the test on ROOMS dataset. Here are some examples of [segmentation output](https://images2.imgbox.com/e0/09/lCcPm3va_o.png). The ARI-fg is 0.377 and mean IoU is 0.340.
> - (6) How to deal with texture-less regions? This is a great question. On ROOMS dataset which has more homogeneous regions, we found that including additional regularization (total variance loss) to encourage smoothness of the segmentation masks helps the masks to cover the entire objects (our model works without this regularization too but the masks become less complete).
> - (7) Thanks a lot for the suggestion from the depth estimation literature! Although depth estimation is not the major focus of our current work, we will incorporate some of these in our future models built on our current work. We will update the reference and discussion.

---

> > ### Comment · Reviewer_31jS · 2022-11-29
> > **Response to author**
> >
> > I am happy that the authors find the depth estimation literature useful.
> > I appreciate the authors' efforts in providing more results in such a short time.
> > Although the new results strongly demonstrate the superiority of the proposed method over previous literature, I am afraid that I have to keep my previous rating.
> >
> > The rebuttal provided by the author is a complete (but good and necessary) overhaul of the experiment section.
> > Such huge changes make it hard for me to evaluate the original paper.
> > To be quite frank, it is the first time that I read a paper with experiment results "based on an early checkpoint of an earlier version of the model".
> >
> > I understand that the proposed method achieves higher scores compared to previous literature already.
> > Judging by the figures shown in the paper, I still believe that the performance can be improved with more tweaks adapted from depth estimation, optical flow, or scene flow literature.
> > If cannot, the author may explain in detail the actual difficulties.

---

> > > ### Author Response · Authors · 2022-12-13
> > > **Thanks**
> > >
> > > Thank you for the feedback. We don't want to make excuse but what happened was that we have reformatted the code to prepare for sharing, and the updated scores were based on training with our new code.
> > >
> > > After reflecting on the suggestion by the reviewer about tweaks from the depth estimation community, we reallize that some of the suggestions actually are already incorporated in a similar way in our model.
> > >
> > > For example, "Bidirectional wrapping": during training, we randomly flip the order of the three frames so that we could use frames t+1 and t to predict frame t-1 (by warping from frame t with estimated object motion from frame t+1 to frame t).
> > >
> > > "Refine the mask across multiple transitions": because the way we sampled triplets (We rendered 7 frames in each scene; we draw triplets of images with equal number of steps between frames for training. This forms 5 triplets with a step of 1, 3 triplets with a step of 2 and 1 triplet with a step of 3 from each 7 frames in a scene), several frames were included in multiple triplets with different numbers of transitions. If we understood correctly, this may be similar to what the reviewer has suggested.
> > >
> > > "Encourage near regions to have similar motion (to reduce over-segmentation within each object)": we actually have not found over-splitting an object to two masks to be a problem for our model, potentially because the model uses coherent motion of all pixels on an object as a cue to learn how to group pixels to an object. This is similar to AMD but we consider 3D motion. However, we agree that this tweak might help fill the holes in the segmentation masks of big objects with coherent colors. And we found that for ROOM dataset with more coherent colors on objects, encouraging adjacent pixels to belong to the same object or background indeed improves performance.

---

### Official Review · Reviewer_BsMb · 2022-10-23

**Confidence:** 4
**Correctness:** 3
**Technical Novelty And Significance:** 2
**Empirical Novelty And Significance:** 2
**Recommendation:** 3

**Clarity, Quality, Novelty And Reproducibility:**

I think the paper is relatively clear and well-written. Many in-line equations make it difficult to follow the text sometimes but overall it is not a huge problem I think. In terms of novelty, given that other work has used a similar warping idea to predict images ([1]), the novelty is limited but still I think the overall technique is original.


**Strength And Weaknesses:**

Overall, as I have said last time, I think this is an interesting paper. The optical flow based warping to predict some subset of the pixels in the next timestep is a nice idea and is perhaps the main novelty of this paper. (However, some recent work has used a similar idea [1].)

I think the writing has improved significantly in the newer version; I found the txt much easier to follow (of course this probably has something to do with the fact that I'm reading paper for a second time.)

I think the main concern with the paper, again like last time, is limited experimental evaluation. The model is evaluated only on a single, rather simple dataset (that was generated by the authors). I would at least have expected a comparison on some more or less standard datasets like CLEVR etc., at least on some of the datasets that other competing models were evaluated on. I understand that the dataset generated by authors have textures, which other techniques find difficult to handle, and their technique can deal with better. But I'd like to see that the proposed technique still works in the "simpler" setting that competing methods can handle.

Other notes

- There are better models than MONet and Slot Attention in this literature like PARTS, SLATE etc. It'd be nice to compare to these as well.
- The authors mention O3V as a similar technique. Why don't you compare against it as well?
- Did the authors think about applying this technique to longer sequences? Longer sequences may make it more challenging to apply the technique but will enable model to learn since longer sequences should have more information.
- There are no std deviations in table 1, which makes it difficult to how whether the performance differences are meaningful.
- Slot attn 128 does worse than smaller slot attn. Do the authors why this is the case? I'd have expected to work at least as well. However, in terms of IoU, it does better. Again, having std dev. here would help to understand what is going on.
- Typos:
  - pg 4., top: modificaiton
  - pg 5., bottom: We -> we
  - pg 8., top: ground-true -> ground-truth

[1] The Emergence of Objectness: Learning Zero-Shot Segmentation from Videos, https://arxiv.org/abs/2111.06394

**Summary Of The Paper:**

Note I have reviewed an earlier version of this paper for last year's conference. I have copied some of the notes from my earlier review but I have read the new version carefully and made sure that I only copied parts that hasn't changed in the new version. Overall, the earlier paper and the current version overlap quite significantly (especially in terms of experimental results).

This paper presents an unsupervised object-centric scene representation technique that can decompose a scene into multiple objects (and segment the scene) and infer their 3D locations and pose. The overall setup is very similar to earlier models like MONet but this model works on sequences of images, more precisely on 3 consecutive images. It uses the first two images to infer the 3D position and pose of objects and combining this with known camera motion tries to predict the last (third) image. Then it uses an optical flow based method to warp the image at time t using the predicted object location/pose/depth to predict (some of) the pixels in image at time t+1. The model is trained to minimize the difference between this prediction and the true next image.

In more detail, the object extraction network outputs the location and pose of each object. And a separate depth perception network outputs the depth for each pixel in the image. The location and pose of objects are used to estimate the velocity of each object (e.g., by subtracting the position at t-1 from position at t. note this requires matching each object at time t-1 to object in time t, which they do using a soft-matching approach). These along with the depth information are then used to warp the image at t to predict pixels in image at time t+1. This is possible only for a subset of the pixels so for the rest, they use a separate "imagination" network that takes in object information and predicts the color/depth and object masks at t+1. The predictions from warping and imagination network are then combined to form the final predicted color and depth images.

To train the model, they require images and camera motion, and use a combination of losses: reconstruction loss on predicted and ground truth image, self-supervised losses on object location, pose, and depth.


**Summary Of The Review:**

Overall, I think the paper is interesting, but unfortunately the empirical evaluation is very limited and this makes it difficult to evaluate the full merit of the proposed approach.

---

> ### Author Response · Authors · 2022-11-25
> **More benchmarking demonstrates superior performance**
>
> Thanks for your suggestion and spending time to review this paper again. We have taken the suggestion to test our model on ROOMS (GQN) data as well. Indeed, the model also works on “simpler” dataset with more homogenous colors (see this [figure for examples of segmentation output on ROOMS](https://images2.imgbox.com/e0/09/lCcPm3va_o.png)). The ARI-fg is 0.377 and mean IoU is 0.340. The replies to other notes are below:
>
> - (1) More benchmarking: we have further tested SLATE and AMD (the reference 1 in your comment). Unfortunately they both fail to place their masks on objects in our dataset. Here are some [examples of segmentation output](https://images2.imgbox.com/b0/80/KGNQl9Bj_o.png). SLATE focused on segmenting backgrounds while AMD treated border segments of the frames as objects. We postulate this is because the models each exploit different types of data distribution. SLATE models spatial dependency of different parts in images and learns a set of dictionaries of image parts. This would work best when images share a common general structure (such as faces) but compositionally vary in each part. Unfortunately such compositionality and common general structure exist mainly in the background rather than the object in our data (objects appear at random locations). AMD might learn better if object pixels move more than the background (in other words, when the self motion is small), but in our data, camera motion is comparable to object motion but camera motion causes large shifts of pixels close to the borders of the image. This may have encouraged the network to model the segmentation flow of the border regions of the image.
> - (2) We have also trained o3v on our dataset but have not made it succeed in generating clear masks on objects in our dataset. We will report further investigation on this soon.
>  - (3) Applying our method to longer sequences is indeed a very interesting idea. We are interested in learning trajectory dynamics associated with objects in our extension. In the current work, we have used different gaps of time intervals to sample the frames collected in each dataset. In our dataset, we included triplets that are equally spaced by 1 step, 2 step, or 3 steps in the simulation. In Carla dataset (in the appendix), we empirically found that using logarithmically spaced gaps between frames up to 16 steps (equivalent to 0.8 s in real world) performs better than using linearly spaced gaps (up to 3 steps) to sample the triplets of frames. We reason this is because longer gaps generate larger optical flow and increased prediction error if the model produces wrong segmentation or object location.
> - (4) No standard deviation: due to the limited GPU resource (the bigger slot attention was trained for 15 days, for example) we have not trained each model for many times to obtain a standard deviation due to different network initiations. But we will report standard deviation for our models soon. To help understand the variation coming from images: the standard error of mean of the scores across objects is below 0.005 in general. The scores were evaluated on the same 4000 images across models.
> - (5) Two versions of slot attention: we think the two metrics are not perfectly correlated. By visual inspection, the smaller slot attention network often segments the background into more slots than the bigger version. Therefore, we subjectively consider slot-attention-128 to be slightly better than the smaller version in the face of discrepant scores. We also found the scores of OPPLE was outdated (early checkpoint of an earlier version of the model) and here are the updated numbers:
> |  Model | ARI-fg | IoU |
> | :---- | :--- |:---|
> | MONet | 0.31| 0.08|
> | MONet-128 | 0.33| 0.22  |
> | MONet-128-bigger| 0.36| 0.20|
> | slot-attention| 0.40| 0.34|
> | slot-attention-128| 0.34| 0.38|
> | SLATE| 0.30| 0.20|
> | AMD| 0.19| 0.02|
> | our model (OPPLE)| **0.65**| **0.47**|
>
> And thanks a lot for pointing out the typos! We really appreciate your careful reading.

---

### Official Review · Reviewer_7oBh · 2022-10-25

**Confidence:** 2
**Correctness:** 3
**Technical Novelty And Significance:** 2
**Empirical Novelty And Significance:** 2
**Recommendation:** 5

**Clarity, Quality, Novelty And Reproducibility:**

The method is clearly presented and written but the key contribution and takeaways are a bit unclear.
Quality: the method shows some improvement over baselines on their proposed dataset and it is understandable. see above
Novelty: Some module designs are sort of novel e.g. jointly infer depth, matching process between two object sets.  But unfortunately, I have a hard time summarizing "what I've learned from the paper".  details see above.
The author promise to release code for reproducibility.

**Strength And Weaknesses:**

Strength:
+ The model use motion cues (wo explicit supervision) to supervise the object discovery tasks and the object notion emerge via bottle-neck object representation.
+ The only supervision is RGB images and camera pose and it is interesting to see depth emerging from unsupervised training.
+ the paper is written clearly and is easy to follow.

Weakness:
- The biggest claimed contribution is that they combine predictive learning with explicit modeling 3D motion. However, the idea of explicitly modeling 3D is not extremely novel e.g. (Henderson & Lamp NeurIPS 2020, Anciukevicius et al NeurIPS 2022). I do not see a fundamental difference between them.  Although [A] Henderson & Lampert, NeurIPS 20 cannot generalize to a single frame, I wonder what happens if both models are inferred from videos – it may favor [A] as OPPLE is a by-frame prediction, yet it helps to understand what is new in OPPLE.
Another important basleines to claim the contribution is to model motions in 2D instead of 3D, but still, use predictive learning to train.
- More ablations are needed to understand how each component contributes to the tasks. Many designs of the method appear rather arbitrary to me, e.g. imagination network chooses not to include geometry even though the warping network does, the effect of regularization terms in training.
- The comparisons are shown on the synthetic dataset generated by their own, which looks like a variant of ROOM datasets with more diverse textures and a fixed number of objects. However, I wonder why ROOM/CLEVER benchmark cannot be evaluated. Although the baselines will improve bc of less diverse textures, according to the authors, the proposed method should also work on that dataset. I encourage authors to report results on more widely adopted datasets in addition to their own datasets. This would calibrate their method better.
- The proposed method does not handle various number of objects like the baselines do.


**Summary Of The Paper:**

This paper learns to discover objects by video prediction. The method called OPPOLE extracts each frame into object states(inferred object 3D position and 1D pose), and background. The method learns the decomposition in an unsupervised method that renders the object representation into image frames and compares it with GT RGB images and inferred depth. Several additional regularization terms are proposed to avoid the degenerate solutions. The method is compared with per-frame unsupervised object discovery baselines including slot-attention and monet on their proposed synthetic room dataset.


**Summary Of The Review:**

Overall, the method is clearly presented and written but the key contribution and takeaways are a bit unclear. The empirical results are relatively weak in the sense that 1) the results are conducted on their own datasets and 2) more ablations need to be done in order to understand each components.

---

> ### Author Response · Authors · 2022-11-25
> **New benchmarking performed**
>
> Thanks for your great suggestions. We reply about the weakness below:
>
> * (1) Contribution: very few works achieved all the goals we aimed for (learning all of depth perception, 3D localization and object segmentation together) with self-supervised learning based only on RGB images and knowledge of self-motion. Indeed Henderson & Lamp (O3V) appears to achieve all of the three with similar constraints as our model. In addition to being able to infer from single images, another advantage of OPPLE is simplicity. By using U-nets for depth perception and segmentation, we do not need a rendering process that may be computationally expensive. We have trained O3V on our dataset but thus far it has not generated reasonable segmentation results; we are investigating the possible cause and will update in a few days. For 2D motion-based models, we now test the appearance-motion decomposition (AMD) model by Liu et al., (NeurIPS 2021) on our dataset. AMD worked on a set of youtube videos but failed on our dataset (see this [figure](https://images2.imgbox.com/b0/80/KGNQl9Bj_o.png) for its example output). One postulated cause is that the model may require objects to have more 2D motion than background to work, as is likely the case in the youtube dataset. But in our data, background pixels can move far due to camera motion, which may lead the model to segment borders of the frame as “objects”.
>
> * (2) Ablation: all the three networks are critical within our framework in order to form a prediction of future images in the presence of occlusion. Empirically, we found that the negative contrastive loss term on different objects’ locations prevents a trivial failure mode as described in the paper. We have prioritized benchmarking additional models after receiving the reviews but will update about the results of the ablation study soon. In terms of the imagination network, it does take geometry information (depth) as input for each individual object/background (masked by the segmentation masks generated by the segmentation network), and its output includes the predicted depth and mask for each object and background. It processes the information of each object and background separately. Therefore, the discrete nature of objects and the 3D information are both captured by the imagination network. Perhaps the geometry you have in mind is a form of rendering from implicit 3D representation as in O3V or NeRF. However, our usage of U-net for imagination makes the prediction process simpler than these models.
>
> * (3) Evaluating on simpler dataset: thanks for the suggestion. We have now added the evaluation on ROOMS dataset (see this [figure](https://images2.imgbox.com/e0/09/lCcPm3va_o.png) for examples of segmentation masks). Our model indeed works well on objects with simple colors too. We found that adding an additional regularization of total variance loss on each object mask further helps the object masks become homogeneous throughout the objects (it works without this regularization too). This regularization compensates for the lack of texture in the ROOMS dataset which is helpful for learning depth and segmentation.
>
> * (4) Handling of various numbers of objects: this is not true. Even though in each scene we placed 3 objects, not all were visible by the camera due to the random placement of the camera. The model can successfully output no segmentation masks when there is no object in the scene. The recurrent neural network inside the segmentation network also allows it to run for longer and output more masks in principle, although we have not tested this. Recurrent network only limits the number of objects by the number of recurrence steps, which is adjustable without the need of changing network architecture. In contrast, attention-based models limit the number of objects by the attention nodes, which is a hard limit.
>
> * (5) To help highlight “what we hope to tell the readers”: we make a concrete step to demonstrate the plausibility of learning rich object-centric representation with similar constraints faced by the brain, with the predictive learning principle. Unlike several other works that also learn 3D object-centric representation, it removes the need of ray-tracing process during learning and inference, and uses simple modified U-nets to infer objects and depth at test time. The updated benchmarking result (printed in other replies), has further demonstrated that the inductive bias adopted by some existing works might have exploited the unique data distribution in certain datasets. But to develop models applicable in more diverse environments, we need to consider the fundamental function of “objectness”. In our perspective, “objects” serve as latent causes of environments which provide abstract handles for making prediction and interaction. We suggest that developing learning principles driven by the functionality of objects may prove fruitful, as it is a cognitive construct developed by the brain through evolution.

---

### Official Review · Reviewer_GkH5 · 2022-10-25

**Confidence:** 4
**Correctness:** 3
**Technical Novelty And Significance:** 3
**Empirical Novelty And Significance:** 2
**Recommendation:** 3

**Clarity, Quality, Novelty And Reproducibility:**

- Clarity: can be improved.
- Quality: results do not seem significant, and the paper fails to discuss a few related works.
- Novelty: the problem setting seems interesting but not entirely novel. The paper fails to highlight its novelty compared to related work.
- Reproducibility: the paper provides a lot of technical detail, but could be more organized.

**Strength And Weaknesses:**

- Strengths
    - The problem setting is interesting and close to how humans perceive objects.
    - The custom dataset seems challenging and can be valuable to the community.
- Weaknesses
    - The empirical results are a bit discouraging. The proposed method (with access to camera intrinsics and multiple frames during training) is similar to or only slightly better than SlotAttention (with access to only static images) in terms of segmentation performance. The benefit of predicting object movement is not well demonstrated by these results.
    - It seems the paper aims to tackle (1) complex object texture and (2) object movement in 3D. However, the paper fails to compare with or discuss the related work in these two aspects.
        - Some recent work (e.g., [GENESIS-v2](https://proceedings.neurips.cc/paper/2021/hash/43ec517d68b6edd3015b3edc9a11367b-Abstract.html) and [SLATE](https://openreview.net/forum?id=h0OYV0We3oh)) tackles complex texture from single 2D images without predicting object movement. The authors did try to train GENESIS-v2 on their dataset, but failed to get good results. Perhaps the authors could try to make some version of the dataset that is similar to those used in GENESIS-v2, and show that predicting object movement leads to better segmentation than using static images alone.
        - Quite a few papers tackle object-centric 3D representations (e.g., [O3V](https://proceedings.neurips.cc/paper/2020/hash/20125fd9b2d43e340a35fb0278da235d-Abstract.html), [ROOTS](https://www.jmlr.org/papers/v22/20-1176.html), [Crawford & Pineau](https://github.com/oolworkshop/oolworkshop.github.io/blob/master/pdf/OOL_19.pdf), [ObSuRF](https://arxiv.org/abs/2104.01148), etc). In principle, these methods could be applied to the dataset in this paper. In fact, the representations they learn are more comprehensive than this paper (because they capture 3D appearance) while given the same input (2D image + camera parameters).
        - While not strictly necessary, the paper could also discuss object-centric representation learning from 2D videos (e.g., [SQAIR](https://arxiv.org/abs/1806.01794), [SCALOR](https://arxiv.org/abs/1910.02384), etc). They treat object movement in 2D, but the representation learning also benefits from predicting future frames.
    - The presentation of the paper can be improved. The paper describes the method at very detailed level, without explaining the high-level intuition/motivation behind the key design choices, making it hard for the reader to follow. In particular, I have the following questions:
        - Why use discrete yaw angles instead of continuous values?
        - Why does the imagination network take original image as input rather than the inferred object latents (mentioned in Section 2.3)?
        - Why use a Von Mises prior distribution for angular velocity?
        - When doing the soft-matching, why do you only use a subset of the latent instead of the full latent?
        - Background velocity is set to zero (mentioned in Section 2.4.1). Do you assume static background?

**Summary Of The Paper:**

The paper presents a method for learning object-centric representations from single 2D images without supervision. Unlike previous work that learns by reconstructing the current frame, the proposed method learns by predicting object movement in future frames. Specifically, the model is trained from triplets of 2D images that are taken consecutively from a 3D scene by a moving camera. The camera intrinsics and movement are known to the model. At test time, the model can be applied to single images for object segmentation and localization. The experiments are done on a custom dataset featuring complex object texture and background. The model shows comparable or slightly better segmentation performance than SlotAttention.

**Summary Of The Review:**

I recommend reject for the paper in its current form, due to potentially limited significance, insufficient discussion of related work, and clarity issues.

---

> ### Author Response · Authors · 2022-11-25
> **Updated performance scores and enriched benchmarking demonstrate the advantage of our model**
>
> Thanks for your great comments. We reply below:
>
> - (1) Discouraging empirical results: after re-examining the result, we found that the previously reported scores of our model were based on an early checkpoint of an earlier version of the model. After updating the scores, our model wins by a large margin (authors have verified the evaluation code is correct and consistently applied to all models):
> |  Model | ARI-fg | IoU |
> | :---- | :---- |:--- |
> | MONet | 0.31| 0.08|
> | MONet-128 | 0.33| 0.22  |
> | MONet-128-bigger| 0.36| 0.20|
> | slot-attention| 0.40| 0.34|
> | slot-attention-128| 0.34| 0.38|
> | SLATE| 0.30| 0.20|
> | AMD| 0.19| 0.02|
> | our model (OPPLE)| **0.65**| **0.47**|
> We wish to emphasize that our goal is not just object segmentation but to learn as many properties of objects as humans can without supervision. Very few models simultaneously achieve all the goals with the same constraints: 3D object localization, depth perception, inference from single images, not using any ground-truth information beyond RGB images or self-motion at training.
>
> - (2) Lack of comparison to previous models: Thanks for the suggestion. As in the table above, we have added comparisons to more models but they underperform on our dataset. Examples of the segmentation results can be seen in this [figure](https://images2.imgbox.com/b0/80/KGNQl9Bj_o.png). Interestingly SLATE focused on segmenting the background while ignoring objects. This may be because in our dataset textures of the background are randomized, which constitutes the major variation in RGB values across scenes and encourages SLATE to prioritize encoding these variations with its dictionaries.
>
> - (3) Other object-centric models with 3D representation: the cited models indeed also learn 3D representation, in particular, by rendering the learned implicit representation at different camera positions. We note that our model does learn 3D representation including 3D appearance, as the imagination network can fill in the part of images unpredictable by warping when predicting the scene at a new camera position/angle (it predicts both the depth and RGB). One general advantage of our model is that it infers depth with one pass of a U-net without the need for an expensive ray tracing process. Additional advantages: O3V and ROOTS both rely on the whole video (or multiple views) to infer objects from the scene, thus cannot easily generalize to segmenting objects from single images, while OPPLE can. Crawford&Pineau and Obsurf appear to be able to infer objects from single images, but have only been tested on environments with identical background and coherent colors on objects and backgrounds. Thus far, O3V has not succeeded in detecting objects from our dataset, but we are investigating the cause and will update soon.
>
> - (4) Add discussion of SQAIR and SCALOR: Thanks for the suggestion. We have added this (unfortunately the updated manuscript was not uploaded properly).
>
> Clarification for the questions:
>
> * (1) Discrete vs. continuous yaw angles: we chose to represent yaw by probabilities over discrete bins because when angles are represented as a single value between -pi and pi, the discontinuity when the pose crosses pi to -pi might cause difficulty for passing the gradient properly in back-propagation. But there may be ways to solve this problem.
> * (2) The imagination network takes images instead of latent as input: your suggestion should work too. Because imagination requires learning statistics in visual scenes (e.g., what often appears on the other side of a car given its appearance on the front), feeding only latent to the imagination network requires the latent to capture more details of the objects. We chose the current approach mainly to encourage the latent code to express a more abstract representation of objects.
> * (3) Von mises distribution as a prior for angular velocity: this simply reflects a prior that in the real world, slower or no motion is more frequent. Practically, we introduce this prior to resolve the ambiguity of angular velocity for objects such as cubes, because a cube with homogeneous color may appear the same after rotating for 15, 105, 195 or 185 degrees. The brain also adopts slow-motion prior.
> * (4) Using a subset of latent code for soft-matching: we think that the latent code should encode more than the identity (and possibly location) of objects in order for the decoder part of the segmentation network to predict object masks, and such features are not necessarily view-invariant. Therefore, we chose this way to allow different features to serve different purposes.
> * (5) Do we assume a static background? Yes, the background does not move relative to the earth (more specifically, the earth reference frame centered at the camera). However, we do randomly draw background texture across scenes and the camera does move within each scene, causing optical flow on the background.

---

> > ### Comment · Reviewer_GkH5 · 2022-12-08
> > **Response to authors**
> >
> > I appreciate the authors' detailed response and added experiments. However, the changes seem too large that the paper probably needs another review iteration.

---

### Author Response · Authors · 2022-12-13
**Additional update after our initial replies**

We thank all reviewers again for the feedback and respect your decisions. Overall, after comparing to several additional models, it appears that only O3V might perform similarly as our model in terms of segmenting images, followed by slot attention. By design, very few models achieve all the three goals we achieved: object segmentation, 3D localization and depth perception. And our model does not require ground-truth information required by some models, such as depth, optical flow, cente-of-mass of objects.

Following our initial replies to each reviewer individually, we have evaluated O3V and Obsurf on our dataset. Qualitatively, O3V learns to segment objects similarly as our model, as shown in this [figure](https://images2.imgbox.com/79/bb/dyts5EjK_o.png). One possible limitation of O3V is it may be sensitive to the assumption of the size of room, as it fails to segment objects when we double the spatial range of possible object locations from the true room size, as shown in this [figure](https://images2.imgbox.com/54/76/1Iy29N9R_o.png). Our model does not strictly depend on the exact information of room size or the absolute location of the camera in the room. We were not able to draw a quantitative conclusion of segmentation performance yet due to the difference in frameworks (tf vs. torch).

We trained both the 2D and 3D versions of Obsurf on our dataset (for the 3D version, we generated a new dataset with the same properties but without object motion because Obsurf does not assume motion of objects, and ground-truth depth was provided as additional input to Obsurf). Visual inspection suggests that it does not perform well on our dataset, as shown in these [figures](https://images2.imgbox.com/8c/89/iiS3piRo_o.png) (the model code could only visualize the segmentation mask in half the resolution on our GPU, although the training does not suffer from this limitation as it samples points from the original images to train). We have not completed quantitative evaluation of segmentation performance yet. But we think it should be consistent with the visual impression.

We also conducted ablation experiments on our model, as suggested, to observe the behavior of the model without using the warping module for prediction. In such cases, the final output of the model depends only on the imagination network. We found that without warping, the model is able to make blurry predictions of a new scene as the camera moves in the scene by learning statistical regularity across the scenes. However, without the warping module, the model does not receive enough teaching signal to learn depth perception or explicit object segmentation maps. This suggests the warping-based prediction is a critical component for the success of the model.

---

### Decision · Program_Chairs · 2023-01-20

**Decision:**

Reject

**Justification For Why Not Higher Score:**

The experiments are lacking, there were too many choices and changes made that are left unexplained and the type of datasets investigated are limited. Papers need major revision and review before it is ready for publication.

**Justification For Why Not Lower Score:**

N/A

**Metareview: Summary, Strengths And Weaknesses:**

The key idea in the paper is to disentangle objects from videos by proposing a temporally structured reconstruction loss that leverages known camera extrinsics and predicted flow/depth. The inductive bias of spatio temporal smoothness is used to predict next frame given past two frames. Experiments are evaluated on synthetic datasets.

Although the problem is quite interesting and unsolved in general, the paper needs a lot more work before it is ready for publication. There was consensus amongst reviewers that the experimental validation is limited. During the rebuttal process there were major revisions to some of the empirical numbers (.e.g IOU scores) without much context and explanation -- it needs further review iterations and checking to ensure that all experiments are sound and valid.